# Autopsy Study Defines Composition and Dynamics of the HIV-1 Reservoir after Allogeneic Hematopoietic Stem Cell Transplantation with CCR5Δ32/Δ32 Donor Cells

**DOI:** 10.3390/v14092069

**Published:** 2022-09-17

**Authors:** Laura E. P. Huyveneers, Anke Bruns, Arjen Stam, Pauline Ellerbroek, Dorien de Jong, Noémi A. Nagy, Stephanie B. H. Gumbs, Kiki Tesselaar, Kobus Bosman, Maria Salgado, Gero Hütter, Lodewijk A. A. Brosens, Mi Kwon, Jose Diez Martin, Jan T. M. van der Meer, Theun M. de Kort, Asier Sáez-Cirión, Julian Schulze zur Wiesch, Jaap Jan Boelens, Javier Martinez-Picado, Jürgen H. E. Kuball, Annemarie M. J. Wensing, Monique Nijhuis

**Affiliations:** 1Department of Medical Microbiology, University Medical Center Utrecht (UMCU), 3584 CX Utrecht, The Netherlands; 2Department of Internal Medicine and Infectious Diseases, University Medical Center Utrecht (UMCU), 3584 CX Utrecht, The Netherlands; 3Department of Hematology, University Medical Center Utrecht (UMCU), 3584 CX Utrecht, The Netherlands; 4Central Laboratory of Translational Immunology, University Medical Center Utrecht (UMCU), 3584 CX Utrecht, The Netherlands; 5IrsiCaixa AIDS Research Institute and Institute for Health Science Research Germans Trias i Pujol (IGTP), 08916 Badalona, Spain; 6Consorcio Centro de Investigación Biomédica en Red de Enfermedades Infecciosas (CIBERINFEC), Instituto de Salud Carlos III, 28029 Madrid, Spain; 7DKMS CC, 72072 Tübingen, Germany; 8Department of Pathology, University Medical Center Utrecht (UMCU), 3584 CX Utrecht, The Netherlands; 9Department of Hematology, Hospital General Universitario Gregorio Maranon, 28007 Madrid, Spain; 10Department of Internal Medicine, Amsterdam UMC, 1105 AZ Amsterdam, The Netherlands; 11Department of Anatomy and Embryology, Leiden University Medical Center, 2333 ZA Leiden, The Netherlands; 12HIV Inflammation and Persistence, Institut Pasteur, Université Paris Cité, 75015 Paris, France; 13Department of Internal Medicine, UMC Hamburg-Eppendorf, 20251 Hamburg, Germany; 14Department of Hematology and Oncology, Princess Maxima Center, 3584 CS Utrecht, The Netherlands; 15Stem Cell Transplant and Cellular Therapies, Pediatrics, Memorial Sloan Kettering Cancer Center, New York, NY 10065, USA; 16Catalan Institution for Research and Advanced Studies (ICREA), 08010 Barcelona, Spain; 17University of Vic—Central University of Catalonia (UVic—UCC), 08500 Vic, Spain

**Keywords:** HIV-1, HIV persistence, reservoir, cure, tissue, CCR5Δ32, allo-HSCT

## Abstract

Allo-HSCT with CCR5Δ32/Δ32 donor cells is the only curative HIV-1 intervention. We investigated the impact of allo-HSCT on the viral reservoir in PBMCs and post-mortem tissue in two patients. IciS-05 and IciS-11 both received a CCR5Δ32/Δ32 allo-HSCT. Before allo-HSCT, ultrasensitive HIV-1 RNA quantification; HIV-1-DNA quantification; co-receptor tropism analysis; deep-sequencing and viral characterization in PBMCs and bone marrow; and post-allo-HSCT, ultrasensitive RNA and HIV-1-DNA quantification were performed. Proviral quantification, deep sequencing, and viral characterization were done in post-mortem tissue samples. Both patients harbored subtype B CCR5-tropic HIV-1 as determined genotypically and functionally by virus culture. Pre-allo-HSCT, HIV-1-DNA could be detected in both patients in bone marrow, PBMCs, and T-cell subsets. Chimerism correlated with detectable HIV-1-DNA LTR copies in cells and tissues. Post-mortem analysis of IciS-05 revealed proviral DNA in all tissue biopsies, but not in PBMCs. In patient IciS-11, who was transplanted twice, no HIV-1-DNA could be detected in PBMCs at the time of death, whereas HIV-1-DNA was detectable in the lymph node. In conclusion, shortly after CCR5Δ32/Δ32, allo-HSCT HIV-1-DNA became undetectable in PBMCs. However, HIV-1-DNA variants identical to those present before transplantation persisted in post-mortem-obtained tissues, indicating that these tissues play an important role as viral reservoirs.

## 1. Introduction

Although antiretroviral therapy (ART) successfully averts HIV-1-related disease progression and saves lives of millions of HIV-infected individuals, the ongoing inflammation and costs urgently call for curative strategies [1]. Current antiretroviral compounds target several steps in the viral life cycle but do not target the integrated provirus nor suppress HIV-1 transcription and production from the cellular reservoir. This integrated provirus forms a stable viral reservoir and is the major barrier to HIV-1 eradication [1].

Several approaches are being explored to eliminate the viral reservoir and ultimately convert HIV-1 infection into a curable disease. Successful viral remission is known in four cases of treatment interruption after allogeneic hematopoietic stem cell transplantation (allo-HSCT) with homozygous CCR5Δ32 donor cells: “The Berlin patient”, two participants in the IciStem cohort; IciS-36, IciS-19—also named “The London patient” and “The Düsseldorf patient”; and recently in a woman of mixed race in New York. The cells used for transplantation lack expression of surface CCR5, rendering them resistant to infection by CCR5-tropic HIV-1 [2,3,4,5,6,7]. Though no viral rebound was observed in these patients after treatment interruption (ATI), traces of proviral HIV-1-DNA could still be found in some samples [3,5,7,8]. Also, in patients transplanted with CCR5-expressing donor cells (CCR5WT), a profound reduction in the viral reservoir was observed. However, in these patients, therapy interruption resulted in a delayed rebound of viruses comparable to the pre-allo-HSCT PBMC population in “the Boston patients”, but not in “the Minnesota patient” [9,10,11,12].

These data highlight the knowledge gap on how allo-HSCT impacts the viral reservoir and which cellular or anatomical compartments fuel HIV-1 rebound. IciStem is an international collaboration to guide and investigate the potential for HIV-1 cure after allo-HSCT and has compiled the largest cohort of HIV-1-infected individuals receiving allo-HSCT (www.icistem.org, accessed on 15 August 2022). Two IciStem CCR5Δ32 patients (IciS-05 and IciS-11) died after allo-HSCT. Post-mortem biopsies of these patients provided a unique opportunity to broadly investigate the viral reservoir after allo-HSCT in multiple tissues, which would never be feasible in living patients.

### 1.1. Cases

#### 1.1.1. Patient IciS-05

A 38-year-old male presented in 1998 with a CD4^+^ T-cell count of 46 cells/μL whole blood and an HIV-1 viral load (VL) >500,000 RNA copies/mL plasma. Despite several changes in his therapy regimen, full suppression was not reached until 2008 (Table 1). Extensive drug resistance was selected and genotypic analyses of the viral envelope predicted infection with CCR5-tropic virus. In 2011, poor-risk myelodysplastic syndrome (MDS) was diagnosed, necessitating allo-HSCT in May 2012. Because no adult HLA-matched unrelated CCR5Δ32 donor was found, a cord-blood transplant of a homozygous CCR5Δ32 donor (HLA-match 4/6) combined with an infusion of CD34+-cells of an HLA-unmatched haploidentical family donor (CCR5wt) was performed, to aim for HIV-1 cure but also to ensure faster engraftment (Table 1). Using general diagnostic assays, viral load was 20 copies/mL on the day of transplant and remained suppressed (<50 copies/mL) after transplantation. Cord-blood donor chimerism in peripheral blood lymphocytes (PBLs) was 57% on day +16 and 100% on day +36. On day +54, chimerism reduced to 95%, and further down to 85% at day +65. Unfortunately, the patient died of severe sepsis on day +68.

#### 1.1.2. Patient IciS-11

A 38-year-old male was diagnosed with HIV-1 in 1993 with a CD4+ T-cell count of 440 cells/. In March 1996, when CD4+ T-cell count fell below 200 cells/μL whole blood, the patient started dual therapy, quickly followed by triple therapy (Table 1). In December 2014, when VL was suppressed and CD4+ T-cells were around 700 cells/μL, he was diagnosed with poor-risk AML. Genotypic analyses of the viral envelope predicted infection with CCR5-tropic virus.

In April 2015, the patient received allo-HSCT with cells of a 10/10 HLA-matched unrelated donor homozygous for the CCR5Δ32 mutation (Table 1). Before allo-HSCT, CD4 count was 283 cells/μL and plasma VL remained suppressed. Donor chimerism decreased from 60% to 0% between day +27 and +55. On day +41, donor chimerism in bone marrow (BM) was only 18%, further indicating graft failure. On day +71, the patient received a second transplant of a heterozygous CCR5Δ32/WT unrelated donor (HLA-match 10/10); the first donor was not available. A month after the second allo-HSCT, on day +100 after the first allo-HSCT, full donor chimerism was reached. On day +108 after the first allo-HSCT and day +37 after the second allo-HSCT, the patient died due to respiratory failure.

## 2. Materials and Methods

### 2.1. Patients and Patient Material

This study included two IciStem patients. IciS-05 was included after signing a general waiver for the use of body material for future research, according to the regulations of the UMC Utrecht HSCT Biobank (HSCT study number 3440). IciS-11 was included after written informed consent was acquired. Ethical approval by the Institutional Medical Ethics Committee of the UMCU was obtained under protocol number NL53114.041.15. Before allo-HSCT, plasma, serum, PBMCs, and BM were obtained and a leukapheresis was performed. From the leukapheresis, 2–3 × 10^8^ PBMCs were sorted into T_N_ (CD3^+^CD4^+^CD45RO^-^CCR7^+^CD27^+^Fas^-^), T_SCM_ (CD3^+^CD4^+^CD45RO^-^CCR7^+^CD27^+^Fas^+^), T_CM_ (CD3^+^CD4^+^CD45RO^+^CCR7^+^CD27^+^), T_TM_ (CD3^+^CD4^+^CD45RO^+^CCR7^-^CD27^+^Fas^+^) and T_EM_ (CD3^+^CD4^+^CD45RO^+^CCR7^-^CD27^-^Fas^+^). CD4^+^ T-cells were obtained from baseline PBMCs and (CD4^+^-cell Isolation Kit, negative selection) (Miltenyi Biotec). At different time points after, allo-HSCT plasma and PBMCs were obtained. Post-mortem biopsies were obtained the day after death from terminal ileum, lung, liver, and spleen. From patient IciS-11, also brain biopsies were obtained and CD4^+^-cells were isolated from fresh ileum and a mediastinal lymph node.

### 2.2. DNA Isolation

Total DNA was isolated from frozen tissue biopsies using two different methods because of size diffence of the biopsy size and the maximum input in the the DNeasy Blood & Tissue kit (Qiagen, Hilden, Germany). Biopsy 1 was sliced into small pieces and cells were lysed using the MagNALyser (Roche, Basel, Switzerland) and purified using QIAquick (Qiagen, Hilden, Germany). Biopsy 2 was lysed and purified using the DNeasy Blood & Tissue kit (Qiagen, Hilden, Germany). Total DNA from PBMCs or CD4^+^-cells was isolated using the DNeasy Blood & Tissue kit.

### 2.3. Patient and Donor CCR5 Genotype Determination

To determine the CCR5 genotype, 30 ng total cellular DNA was amplified using CCR5forward2 5′-GATAGGTACCTGGCTGTCGTCCAT-3′, and CCR5d 5′-CCTGTGCCTCTTCTTCTCATTTCG-3′. As a control, plasmids containing the wild type CCR5 sequence or the CCR5∆32 sequence were also amplified and lengths of all amplified fragments were compared by gel-electrophoresis and the Bioanalyzer (Agilent, Santa Clara, United States). Deletions in CCR5 were confirmed by sequence analysis.

### 2.4. Ultra-Sensitive Viral Reservoir Quantification

Ultra-sensitive HIV-1 proviral DNA quantification was performed using primers in the conserved HIV-LTR region and pol region and total cell DNA was quantified using an RPP30 (Ribonuclease P/MRP Subunit P30) primer and probe set as described in Bosman et al. [13]. HIV-1 DNA was quantified using de droplet digital PCR (ddPCR, Bio-Rad). The threshold was set manually at a conservative fluorescence level. Water was used as a no-template control and DNA isolated from PBMCs of HIV-negative donors was used as a DNA template control. U1 cells were tested as a positive control. In 3/100 assays, a single positive droplet was observed. Thus, measurement of a single droplet is interpreted as a trace.

### 2.5. Micro-Chimerism Assessment

In addition to routine clinical chimerism testing, ultra-sensitive chimerism PCR was performed on peripheral blood (day +55, +106), post-mortem lymph node, and CD4^+^-cells from the lymph node of patient IciS-11 using the Mentype DIPscreen and Mentype DIPquant assays (Biotype), with a sensitivity of 0.01–0.001%, depending on quality and quantity of the purified DNA.

### 2.6. Ultra-Sensitive Plasma HIV-1 RNA Determination

Ultra-sensitive VL was measured using Cobas Ampliprep/Cobas TaqMan HIV-1 Test v2.0 (Roche Molecular Systems, Inc. Pleasanton, CA, USA) after ultracentrifugation of 4–9 mL of plasma [14].

### 2.7. HIV-1 Co-Receptor Prediction and Determination

Genotypic HIV-1 co-receptor tropism was predicted by deep sequence analysis of the V3 loop of the viral envelope (gp160-V3). Total cellular DNA was amplified in triplicate in a nested approach using primers envF1.1 5′-GGATATAATCAGYYTATGGGA-3′, envF1.2 5′-GAGGATATAATCAGTTTATGG′, envR1.1 5′-GGTGGGTGCTAYTCCYAITG-3′, envR1.2 5′-GGTGGGTGCTATTCCTAATGG-3′ for the first amplification and primers envF2.1 5′-GATCAAAGCCTAAARCCATGT-3′, envF2.2 5′-GATCAAAGCCTAAAGCCATG-3′, envR2.1 5′-CTCCAATTGTCCYTCATHTYTCC-3′, envR2.2 5′-ACTTCTCCAATTGTCCCTCATAT-3′ in the nested amplification. The DNA concentration of the purified amplicons was determined using Quant-iT PicoGreen dsDNA Assay (Thermo Fisher Scientific, Waltham, MA, USA) and a sequence library was formed using the Nextera DNA Library Preparation Kit (Illumina, San Diego, CA, United States). Finally, the DNA concentration of the library was measured (Quant-iT PicoGreen dsDNA Assay Kit, Thermo Fisher Scientific, MA, USA), diluted to 2 ng/mm^3^ denatured, and sequenced using the MiSeq Reagent Kit v2 (500-cycles) (Illumina, San Diego, USA). Viral tropism was predicted in silico using geno2pheno (454) with a 3.5% cut-off indicating the probability of classifying an R5-tropic virus falsely as CXCR4-tropic virus (FPR; false-positive-rate) [15].

Phenotypic HIV-1 co-receptor was determined using cells and virus culture. *Cells:* U373-MAGI-CCR5E, U373-MAGICXCR4CEM [16], MT2 cell lines [17] (NIH AIDS Reagent Program), and SupT1R5X4 (personal gift from J. Hoxie) were maintained as recommended. PBMCs from five healthy donors (homozygous for CCR5WT) were prepared by Ficoll–Paque density gradient centrifugation of heparinized blood. The mix was stimulated for 3 days with phytohaemagglutinin (2 mg/L) in culture medium [RPMI1640 with L-glutamine (BioWhittaker, Lonza, Basel, Switzerland), 10% fetal bovine serum (FBS; Biochrom AG, Berlin, Germany) and 10 mg/L gentamicin (Gibco)]. Virus culture: PBMCs from healthy donors were co-cultured with purified CD4^+^-cells from patients IciS-05 and IciS-11 for 2 h at 37 °C, after which cells were washed twice. Subsequently, cells were cultured with 25 U/mL IL-2 in 10 mL CM. Virus was cultured for 3 weeks, and twice-weekly half of the culture was replaced with fresh culture medium with 25 U/mL recombinant human IL-2 (Thermo Fisher Scientific, Massachusetts, United States) and once weekly freshly stimulated PBMCs from healthy donors were added. Twice weekly HIV-1 replication was monitored by CA-p24 ELISA and culture supernatant was stored for analyses of viral co-receptor tropism in MT2 cells (expressing CD4 and CXCR4) and as a control SupT1R5 × 4 cells (expressing CD4, CCR5 and CXCR4).

Analysis of HIV-1 co-receptor usage in U373-MAGI cells: Infection of these cells was done using 2 ng of CA-p24 of the patient-derived viral isolate or the control viruses [6].

Phylogenetic sequence analysis; viral evolution and compartmentalization: HIV-1 subtype was determined using REGA HIV Subtyping Tool [18]. The evolutionary history was inferred using Maximum Likelihood (MEGA 7) (500 bootstraps) [19]. The tree with the highest log likelihood is shown. Initial tree (s) for the heuristic search were obtained automatically by applying Neighbor-Join and BioNJ algorithms to a matrix of pairwise distances estimated using the Maximum Composite Likelihood (MCL) approach, and then selecting the topology with superior log likelihood value. The tree is drawn to scale, with branch lengths measured in the number of substitutions per site. All positions containing gaps and missing data were eliminated.

## 3. Results

### 3.1. Patient IciS-05

Twenty days before allo-HSCT, only an unquantifiably low amount of HIV-1 RNA could be detected with routine diagnostics. Ultra-sensitive plasma viral load was 15 copies/mL (Figure 1a). The viral reservoir was quantified in BM, PBMCs, naïve T-cells, and all memory T-cell subsets with HIV-1-DNA being most abundant in the more differentiated T-cell subsets (1135-6924 HIV-1-LTR copies/10^6^ cells) (Table 2).

In-depth genotypic analyses of the viral reservoir showed that the population was rather homogeneous with two viral variants dominating all cell types (Appendix A). Genotypic analyses predicted CCR5 co-receptor dependency for both variants (FPR 87.2% and 89.7%) (Table 2). Viruses cultured from PBMCs also showed CCR5-tropism and were unable to infect MAGI-X4 cells (Appendix A) and MT2 cells.

After allo-HSCT with CCR5Δ32 donor cells, a decrease in proviral DNA was observed in PBMCs (Figure 1a, Table 2). On day +36, when full chimerism was observed in PBLs, proviral DNA in PBMCs and plasma VL were both below the limit of quantification. Subsequently, a decrease in chimerism was observed in peripheral T-cells, and HIV-1-DNA in PBMCs and plasma VL both became detectable on day +54. On day +68, when the patient deceased, plasma VL was detectable (8 copies/mL) but proviral DNA in PBMCs was still undetectable. Interestingly, at the same time, proviral DNA could be detected in all post-mortem biopsies (Table 2). Viral envelope could be amplified and sequenced from lung tissue and terminal ileum and revealed the dominance of the same CCR5-tropic viral variants as observed in peripheral blood and BM prior to allo-HSCT (Appendix A).

### 3.2. Patient IciS-11

Shortly before allo-HSCT, low-level plasma HIV-1 RNA was detectable by ultrasensitive viral load assay (2 copies/mL) (Figure 1b). HIV-1-DNA could be detected in BM, PBMCs, naïve T-cells, and all memory T-cell subsets with most HIV-1-LTR DNA in the differentiated subsets (73-4629 HIV-1-LTR copies/106 cells) (Table 3).

While proviral DNA levels were lower in comparison to IciS-05, in-depth genotypic analyses of the viral envelope sequence demonstrated more inter- and intra-sample variation (Table 3, Appendix A). All variants were predicted for the use of the CCR5 co-receptor for viral entry (FPR 9.7–77.3%) (Appendix A). HIV-1 variants cultured from BM and PBMCs were also CCR5-tropic and unable to use the alternative CXCR4 co-receptor as present in MAGI-X4 cells (Appendix A) and MT2 cells.

Shortly after allo-HSCT, plasma HIV-1 RNA could no longer be detected and also PBMC proviral DNA rapidly declined (Table 3, Figure 1b). Over time, with decreasing chimerism, proviral DNA increased to pre-transplantation levels and plasma ultrasensitive VL became detectable (3 copies/mL). On day +55 when the graft was rejected, proviral DNA in PBMCs was twice the pre-transplant level. Considering the relatively low cell counts after allo-HSCT, a relatively big increase in proviral DNA was observed. In-depth sequence analyses revealed the same viral sequences as seen before transplantation, and thus no signs of viral evolution or escape of CCR5 co-receptor usage were observed.

On day +71, IciS-11 received a second allo-HSCT with heterozygous CCR5Δ32/WT donor cells. In the first week after transplantation, a rapid decrease of proviral DNA was observed. On day +35, when chimerism was 100%, PBMC proviral DNA and plasma VL were undetectable. Two days later the patient died of respiratory failure. Multiple post-mortem biopsies were obtained from liver, lung, spleen, brain, and lymph node. In contrast to patient IciS-05, detection of HIV-1-DNA in these tissues varied between the different tissues and biopsies, with lung and brain being negative and other tissues being positive in only one of the biopsies. The viral envelope could not be successfully amplified from any of the biopsies (Table 3).

## 4. Discussion

To date, allo-HSCT is the only strategy to profoundly reduce the HIV-1 reservoir irrespective of whether CCR5 wild-type stem cells or stem cells lacking expression of the CCR5 co-receptor for HIV-1 entry are used [2,3,4,5,7,8,9,11,20,21,22].

Autopsy of patients IciS-05 and IciS-11 provided a unique opportunity to investigate the impact of allo-HSCT on the dynamics and composition of the viral reservoir in PBMCs as well as in different tissue compartments. In patient IciS-05 who was transplanted in a non-myeloablative procedure with CCR5Δ32/Δ32 donor cells, no HIV-1-DNA could be detected by ddPCR in PBMCs but HIV-1-DNA could readily be detected in all biopsies taken from lung, liver, spleen, and ileum. A study describing post-mortem HIV-1 reservoir analysis in a perinatally infected child who also underwent a myeloablative procedure using cord blood CCR5Δ32 donor cells could not detect HIV-1-DNA in blood but could detect viral DNA in some of the tissues analyzed using sensitive in-situ hybridization techniques [21]. The sensitivity of the ddPCR as employed in the study of Rothenberger et al. was likely affected by the smaller number of cells that could be assessed from the tissue biopsies as compared to our study.

In patient IciS-11, who was also transplanted with CCR5Δ32 donor cells, an initial decrease of proviral DNA in PBMCs was followed by an increase in proviral DNA perPBMC. Eberhard et al. showed an HIV-1-specific T-cell activation in IciStem patients after allo-HSCT and the risk of viral escape from the HIV-1 reservoir shortly after transplantation [23]. In-depth sequence analyses of the viral envelope gene revealed identical viral sequences as seen before transplantation of patient IciS-11 and as such showed no signs of viral evolution or escape of CCR5 co-receptor usage. As patient IciS-11 experienced cytomegalovirus (CMV) reactivation after allo-HSCT during incomplete chimerism, we hypothesize that the increase of proviral DNA could be due to the expansion of particular T-cell populations after allo-HSCT. This observation is in line with the described expansion of HIV-1-infected CD4+ T-cells in response to CMV or Epstein–Barr viral antigens, contributing to sustain the HIV-1-DNA reservoir following chemotherapy [12].

Patient IciS-11 did not suffer from GvHD and, like the Berlin patient, was transplanted twice. HIV-1-DNA copy number in the tissues was at the detection level of our ddPCR assays with discrepancies between different tissue biopsies and target regions. These discrepancies show the difficulties analyzing proviral DNA in high-volume post-mortem tissues and emphasize the challenges that we face in the accurate detection and quantification of proviral DNA in low-volume biopsies of patients undergoing curative strategies. Though with a successful transplantation full chimerism in the peripheral blood is reached in the weeks after allo-HSCT, little is known about the turnover and donor chimerism in tissues. In the Berlin patient, CCR5-positive macrophages were still detected in rectal tissue more than 5 months after allo-HSCT [4]. In patient IciS-11, who died 1 month after his second transplant, chimerism in PBLs was 100% and no HIV-1-DNA was detected in blood. Simultaneously, in CD4^+^-cells obtained from the lymph node, where donor chimerism only reached 38%, HIV-1-DNA was detected, most likely in the recipient cells.

In patient IciS-05, proviral DNA from the different tissue samples was genotypically characterized and phylogenetic analysis indicated that all tissue-derived sequences intermingled with pre-transplantation viral populations as present in the different T-cell subsets and BM. These data show no evident compartmentalization between viruses obtained from peripheral blood and BM before the transplant compared to viruses obtained from lung and ileum afterwards. Unlike in the Berlin patient, sequences obtained from the different post-mortem tissues after allo-HSCT showed no evidence of persistence of macrophage-tropic viral variants. In general, our data are in agreement with post-mortem studies demonstrating the lack of compartmentalization of HIV-1 envelope sequences observed in lung, lymph nodes, and colon [24,25]. Previous ATI studies performed after allo-HSCT suggested that the observed viral rebound was caused by pre-existing viral variants present in the pre-transplant PBLs [12,26,27]. However, these patients’ tissues were not characterized. Since no HIV-1-DNA-positive cells could be found in blood, the rebounding viruses post allo-SCT most likely originated from tissue compartments.

A limitation of this study that should be addressed is the short follow-up of the patients after allo-HSCT. Both patients died shortly after allo-HSCT, hampering the follow-up of the dynamics of the donor cells in different tissues. Also because sampling occurred shortly after allo-HSCT, only limited cells were available due to the allo-HSCT procedure. Analyses of the proviral DNA could only be performed on PBMCs and not on a more concentrated CD4+ T-cell subset, which could have underestimated the true size of the peripheral reservoir after allo-HSCT.

In conclusion: This is the first report that performed in-depth post-mortem tissue quantification and genotypic characterization after allo-HSCT with CCR5Δ32/Δ32 donor cells of HIV-1-infected individuals. We demonstrate that the detection of proviral DNA in tissues was prolonged compared to PBMCs. Tissues clearly play an essential role as a long-standing viral reservoir and routine sampling in living HIV-1-individuals will be insufficient to represent the extent of this reservoir. Before conducting ATI, the role of the tissue reservoir has to be taken into consideration.

## Figures and Tables

**Figure 1 viruses-14-02069-f001:**
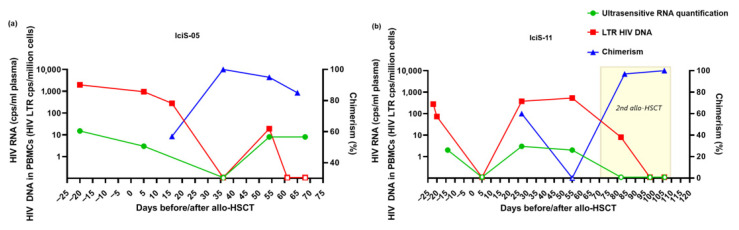
Clinical and virological data of IciS-05 (**a**) and IciS-11 (**b**) pre- and post-allo-HSCT. HIV-1 proviral DNA expressed as LTR DNA copies/million PBMC, and HIV-1 RNA as copies/mL for plasma and chimerism. Open symbols represent values under the level of quantification. Abbreviations: allo-HSCT, allogeneic hematopoietic stem cell transplantation. PBMCs, peripheral blood mononuclear cells.

**Table 1 viruses-14-02069-t001:** Baseline clinical characteristics of patients.

	IciS-05	IciS-11
**Hematological Data**		
Hematological diagnosis	MDS	AML
Donor type/graft source	Cord blood + CD34+-cells third party donor	HLA-matched unrelated donor
Donor CCR5	Homozygous CCR5Δ32	First donor: homozygous CCR5Δ32, second donor: heterozygous CCR5Δ32/WT
Recipient HLA	HLA-A*03:01;24:02; HLA-B*07:02;35:01; HLA-Cw*04:01;07:02; HLA-DRB1*01:01;04:04; HLA-DQB1*03:02;05:01	HLA-A*01:01;02:01; HLA-B*07:02;-; HLA-Cw*07:02;-; HLA-DRB1*15:01;-; HLA-DQB1*06:02; -
Donor-recipient HLA match	4/6 cord blood donor (A*03:01; -; B*07:02; 35:02; C*07:02; 04;01; DRB1*01:01; 11:04); 50% haploidentical family member (A*24:02;26:01;B*07:02; 14:01;C*07:02;08:02; DRB1*04:04; 14:54; DQB1*03:02; 05:03)	10/10; 10/10
Pre-transplant chemotherapy	None	2 induction cycles with cytarabine and idarubicin
Conditioning	ATG, fludarabine with busulvex	ATG, fludarabine, and low-dose TBI before initial transplant; ATG fludarabine and treosulfan before second transplant
GvHD prophylaxis	Cyclosporine, prednisone	Cyclosporine, prednisone, and mycophenolic acid mofetil
GvHD severity	Acute skin GvHD grade 1	No GvHD
**Virological data**		
Time from HIV-1 diagnosis to allo-HSCT	14 years	22 years
Time from start cART to allo-HSCT	14 years	19 years
Host CCR5	CCR5WT/WT	CCR5WT/WT
Predicted HIV-1 co-receptor tropism	CCR5-tropic virus (FPR 68.0–96.2%)	CCR5-tropic virus (FPR 9.7–77.3%)
Phenotypic HIV-1 co-receptor tropism	CCR5-tropic virus	CCR5-tropic virus
HIV-1 subtype	HIV-1 subtype B	HIV-1 subtype B
cART History	1998: AZT/3TC, NFV. 2005: TNF, EFV, ATV/r 2006: TNF, LPV/r, NVP 2006: TNF, SQV/r, NVP 2008: AZT/3TC, TNF, SQV/r	1996: AZT, DDI. 1996: D4T, 3TC, SQV 1997: D4T, 3TC, SQV/r 1999: D4T, 3TC, NVP 2003: TNF, 3TC, NVP
cART during allo-HSCT procedure	Day -152: TNF/FTC, DRV/r, RAL Day -34 until +68: TNF/FTC, RAL, MVC, ENF Day -5 until +37: ETR added	Day -19: TNF, FTC, DTG Day +20 until +107: ABC/3TC, DTG
Plasma HIV RNA load at allo-HSCT	20 copies/ml	<50/TND copies/ml
HCV	Anti-HCV negative	Anti-HCV negative
HBV	HbsAg negative, anti-HBc positive, anti-Hbs negative	HbsAg negative, anti-HBc positive, anti-Hbs positive
CMV status pre-SCT	Positive	Positive
Donor CMV status	Positive	Positive

3TC, lamivudine. ABC, abacavir. Allo-HSCT, allogeneic hematopoietic stem cell transplantation. AML, acute myeloid leukemia. Anti-HBc, antibody against hepatitis B core antigen. Anti-HBs, antibody against hepatitis B surface antigen. ATG, anti-thymocyte globulin. ATV/r, ritonavir boosted atazanavir. AZT, zidovudine. cART, combined antiretroviral therapy. CCR5, C-C chemokine receptor type 5. CCR5Δ32, C-C chemokine receptor type 5 delta 32 mutation. CMV, cytomegalovirus. DDI, didanosine. DRV/r, ritonavir boosted darunavir. DTG, dolutegravir. D4T, stavudine. EBV, Epstein-Barr virus. EBV Vca, EBV viral capsid antigen antibody. EBV EBNA, EBV nuclear antigen antibody. EFV, efavirenz. ENF, enfuvirtide. ETR, etravirine. FTC, emtricitabine. FPR, False Positive Ratio. GvHD, Graft-versus-host disease. HBV, hepatitis B virus. HbsAG, hepatitis B surface antigen. HCV, hepatitis C virus. HLA, human leukocyte antigen. LPV/r, ritonavir boosted lopinavir. MDS, myelodysplastic syndrome. MVC, maraviroc. NFV, nelfivnavir. NVP, nevirapine. RAL, raltegravir. RS, respiratory syncytial virus. RTV or r, ritonavir. SQV/r, ritonavir boosted saquinavir. TBI, total body irradiation. TDF, tenofovir disoproxil fumarate. TND, target not detected. TPHA, Treponema pallidum haemagluttination test. WT, wild type.

**Table 2 viruses-14-02069-t002:** Results of HIV-1-DNA quantification and characterization in patient IciS-05.

Time Point (Days)	Chimerism (%)	Ultrasensitive RNA Quantification (Copies/mL)	Patient Material	ddPCR (Copies/10^6^ Cells)		Gp120V3-Sequence (Dominant FPRs; Range, %)
				HIV-1 LTR	HIV-1 Pol	
Pre allo-HSCT						
−20		15	PBMCs	1967	432	87.2, 89.7 (68.8–96.2)
			Tn	1284	167	87.2, 89.7 (87.2–89.7)
			Tcm	3074	609	87.2, 89.7 (87.2–91.0)
			Ttm	5600	1622	87.2, 89.7 (68.0–89.7)
			Tem	6924	1886	87.2, 89.7 (68.8–89.7)
−19			BM	1135	167	87.2, 89.7(73.7–89.7)
Post allo-HSCT						
+5		3	PBMCs	949	ND	
+16	57		PBMCs	278	<21	
+36	100	0	PBMCs	<5	ND	
+54	95	8	PBMCs	19	ND	
+61			PBMCs	<20	<20	
+65	85					
+68 (died)		8	PBMCs	<7	<22	
Post-mortem biopsies (one biopsy from same site is separated in two parts; 1,2)						
+69			Liver, biopsy 1; 2	54; trace	ND; <8	No amplification
			Lung left, biopsy 1; 2	36; 49	ND; trace	89.7, 87.2 (87.2–89.7)
			Lung right, biopsy 1; 2	90; 32	ND; trace	89.7, 87.2 (71.4–92.2)
			Spleen, biopsy 1; 2	43; 67	ND; 34	No amplification
			Terminal Ileum, biopsy 1; 2	549; 81	ND; 89	89.7, 87.2 (64.4–89.7)

Allo-HSCT, allogeneic hematopoietic stem cell transplantation. BM, bone marrow. ddPCR, droplet digital PCR. FPR, false-positive-rate. LTR, long terminal repeat DNA sequence. ND, not done. NTC, no template control. PBMCs, peripheral blood mononuclear cells. pol, DNA polymerase DNA sequence. Tcm, central memory T-cell. Tem, effector memory T-cell. Tn, naive T-cell. Ttm, transitional memory T-cell.

**Table 3 viruses-14-02069-t003:** Results of HIV-1-DNA quantification and characterization in patient IciS-11.

Time Point (Days)	Chimerism (%)	Ultrasensitive RNA Quantification (Copies/mL)	Patient Material	ddPCR (Copies/10^6^ Cells)		Gp120V3-Sequence (Dominant FPR; Range, %)
				HIV-1 LTR	HIV-1 Pol	
Pre allo-HSCT						
−22			PBMCs	279	21	18.0 (18.0–70.5)
−20			BM	73	<14	70.5 (39.4–70.5)
−14		2	Tn	571	69	31.8 (18.0–70.5)
			Tscm	490	ND	9.7 (9.7–50.3)
			Tcm	2222	544	42.4 (22.1–77.3)
			Ttm	2780	838	42.4 (18.0–70.5)
			Tem	4629	882	21.8 (18.0–39.4)
Post allo-HSCT						
+5		0	PBMCs	trace	<5	
+27	60	3	PBMCs	378	ND	
+55	0	2	PBMCs	534	72	31.8 (31.8–70.5)
Post 2nd allo-HSCT (days post 1st allo-HSCT)						
+11 (+82)		0	PBMCs	8	ND	
+27 (+98)		0	PBMCs	<2	<4	
+35 (+106)	100	0	PBMCs	<2	ND	
Post-mortem biopsies (one biopsy from same site is separated in two parts; 1,2)						
+37 (+108)			Liver, biopsy 1; 2	60; <7	ND; trace	No amplification
			Lung left, biopsy 1; 2	28; <4	ND; <4	No amplification
			Lung right, biopsy 1; 2	trace; <3	ND; <3	No amplification
			Spleen, biopsy 1; 2	60; <6	ND; <6	No amplification
			Brain, biopsy 1; 2	<4; <14	<4; <4	No amplification
	38		LN CD4^+^ cells	10	<4	No amplification

Allo-HSCT, allogeneic hematopoietic stem cell transplantation. BM, bone marrow. ddPCR, droplet digital PCR. FPR, false-positive-rate. LN, lymph node. LTR, long terminal repeat DNA sequence ND, not done. NTC, no template control. PBMCs, peripheral blood mononuclear cells. pol, DNA polymerase DNA sequence. Tcm, central memory T-cell. Tem, effector memory T-cell. Tn, naive T-cell. Tscm, stem memory T-cell. Ttm, transitional memory T-cell.

## Data Availability

The authors confirm that the data supporting the findings of this study are available within the article. Further inquiries can be directed to the corresponding author.

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
