# Peer review of "Autopsy Study Defines Composition and Dynamics of the HIV-1 Reservoir after Allogeneic Hematopoietic Stem Cell Transplantation with CCR5Δ32/Δ32 Donor Cells"

_viruses, 2022, doi:10.3390/v14092069_

Round 1

Reviewer 1 Report

The authors present an in-depth analysis of HIV-1 viral dynamics in two patients who received bone marrow transplants from homozygous CCR5delta32 donors. PBMC and bone marrow samples taken before and after transplantation were studied as well as post-mortem tissue samples. The authors determined the presence of CCR5-tropic viral genomes in all samples analysed before transplantation, however following transplantation no detectable viral genomes were present in PBMC’s. Likewise viral genomes were present in post-mortem tissues but not PBMC’s. The authors conclude that this study highlights the fact that viral genomes present before transplantation with CCR5delta32 stem still persist in tissue reservoirs even when virus is no longer detectable in PBMCs. This work adds to the knowledge required to formulate cure strategies in HIV infected individuals.

 Comments on improving the manuscript:

1 Introduction:

In the introduction and in other places in the document there is inconsistent referring to virus as either HIV or HIV-1. For the purpose of this study HIV-1 should be used throughout the document to not confuse with HIV-2.

Line 64 may be confusing as it seems like there are six examples not four – maybe use brackets to list the two IciStem patients (i.e. IciS-36, “the London patient” and IciS-19, “The Düsseldorf patient”)

Cases:

Line 84 and other places in the document, CD4+ T-cell count is given as being present in plasma. This should be whole blood since plasma is cell free. Also consider using cells/ul since the mm3 should be in superscript format.

Line 92 HLA-match 4/6 – it would be useful for the authors to indicate which HLA loci were a match between donor and recipient, possibly this could be indicated on Table 1.

Line 92 Could it be explained why the patient received both homozygous CCR5delta32 and CCR5wt cells at the same time? Should this fact be included in Table 1?

Table 1 – check that all abbreviations are accounted for in the legend – ATG?

 2 materials and methods:

Line 142 “the day after death

DNA isolation

It is stated in the previous paragraph that tissue biopsy were split in two parts to avoid unequal distribution of provirus when extracting. However, the two biopsies were processed with different extraction methods instead of using the same method in separate extractions. What was the reason for using two different DNA extraction methods? You would not now if genomes were missed because of a difference in extraction method rather than differential compartments in the biopsy.

Ultra-sensitive viral reservoir quantification

Droplet digital PCR was used. This section could be better explained – it may be too brief. This is an important assay used to detect the and quantify viral reservoirs.

Phylogenetic sequence analysis

This method could also be expanded to include a description of the PCR method used to generate a sequence fragment for phylogenetic analysis. The image is not very clear, the sequence labels are also very long – this might be better as a supplementary image.

3. Results

Line 226: copies/106 the 106 should be in superscript format.

In Table 2, in pre-transplantation, PBMC’s and CD4 T-cell subsets were examined for the presence of viral genomes.

Why were subsequent samples post-transplant only tested for viral genomes in PBMC’s? Was there insufficient cell numbers to purify T-cell subsets? This would have impacted on the sensitivity of detecting viral genomes due to a lack of enrichment of CD4+ T cells. This should be mentioned as a limitation of the study.

Line 244: when the patient was deceased

Line 281: is the reason why the same homozygous CCR5delta32 donor could not be used for a second infusion known?

Figure 2: this figure could be better explained in the legend – specifically which gene fragment was used for the tree and why was a consensus C V3 used as a root – should subtype B sequences also be included since this was a subtype B infection? Why was a longer envelope fragment not used – the V3 loop is very short.

Discussion:

In patient IciS-11, who was also transplanted with CCR5Δ32 donor cells, an initial 314 decrease of proviral DNA in PBMCs was followed by a >15-fold increase in proviral DNA 315 per CD4+ T-cell.”

It’s not clear how this fold change was calculated. Should you consider having fold change for proviral DNA in PBMC in tables 2 and 3. Also is “per CD4+ T-cell” an error should this be per 10^6 PBMC’s – CD4 T-cells were not purified and examined separately after transplant?

The discussion should contain a section on limitations of the study. Such as PBMCs vs CD4+ T cell analyses. The patients could not be followed up over a longer period to observe further changes in dynamics or to see an effect of using CCR5delta32 heterozygous cells in the second transfusion. An analysis of defective viral genomes may have also been useful.

 Overall this work clearly presents new data in the stem cell transplant for HIV cure arena and I recommend publication with an attempt to address some of the comments above. Thank you.

Reviewer 2 Report

Although two patients, IsiS-05 and IciS-11, died after CCR5Δ32/Δ32 allo-HCST, this manuscript (viruses-1917284) has valuable information and worth to be published.  

Minor points:

“Post-mortem analysis of IciS-05 revealed proviral DNA in all tissue biopsies, but not in PBMCs.”
Why does HIV DNA exist in all tissue biopsies, but not in PBMC? Could authors give an explanation in the discussion?
